# Evaluation of gene expression of PLEKHS1, AADAC, and CDKN3 as novel genomic markers in gastric carcinoma

**Marwa Sayed Abdel-Tawab**[1]*, **Hanan Fouad**[2,3], **Asmaa M. Othman**[4], **Ragaey A. Eid**[5], **Marwa Abdeltawab Mohammed**[6], **Ahmed Hassan**[7], **Hoda Ramadan Reyad**[1]

1 Medical Biochemistry Department, Faculty of Medicine, Beni-Suef University, Beni-Suef, Egypt,
2 Department of Medical Biochemistry, Faculty of Medicine, Cairo University, Giza, Egypt, 3 Galala University, Faculty of Medicine, Suez Governorate, Suez, Egypt, 4 Department of Internal Medicine, Faculty of Medicine, Beni-Suef University, Beni-Suef, Egypt, 5 Department of Tropical Medicine (Department of Gastroenterology, Hepatology and Endemic Medicine), Faculty of Medicine, Beni-Suef University, Beni-Suef, Egypt, 6 Medical Physiology Department, Faculty of Medicine, Beni-Suef University, Beni-Suef, Egypt, 7 Clinical Oncology Department, Faculty of Medicine, Beni-Suef University, Beni-Suef, Egypt

* drmarwasayed2016@gmail.com, marwa.amar@med.bsu.edu.eg

**Data Availability Statement:** All relevant data are within the manuscript and its Supporting Information files.

## Abstract

Gastric cancer (GC) is considered lethal aggressive cancer. In Egypt, GC has a low incidence but unfortunately, it is mostly diagnosed at an advanced stage with a poor prognosis. Assessment of novel markers that can be used in the early detection of GC is an urgent need. The present study was performed to assess the association of the Pleckstrin homology domain-containing S1 (PLEKHS1), arylacetamide deacetylase (AADAC, and Cyclin-dependent kinase inhibitor 3 (CDKN3) genes with GC and to correlate their gene expression levels with tumor stage, grade, and other clinicopathological features. The current work was performed on forty gastric tissue samples; twenty in Group 1 with GC tissues at different stages, and grades and twenty in Group 2 (control group) with non-tumorous tissue. PLEKHS1, AADAC, and CDKN3 gene expression were assessed by RT-qPCR. AADAC, CDKN3 genes were significantly ($p<0.001$) upregulated, while PLEKHS1 gene was significantly ($p<0.001$) downregulated in the GC group than the control group. AADAC gene expression exhibited a high significant ($p<0.001$) positive correlation with the tumor grades and the tumor stages. A high significant negative correlation between AADAC, and CDKN3 gene expression ($r = -.760$, $p<0.001$) was found. The three studied parameters showed high significant sensitivity and specificity in the prediction of the presence of GC. PLEKHS1, AADAC, and CDKN3 gene expressions were suggested to be used as diagnostic and predictive biomarkers of GC, additionally, AADAC may be used as a prognostic marker in these patients for further future confirming studies.

**Funding:** This research did not receive any specific grant from any funding agency in the public.

## Introduction

Gastric cancer (GC) is the third one among the most common causes of death from cancer [1]. GC is always diagnosed at an advanced stage with a poor prognosis. It was reported that, GC in Egypt is the 12th most common cancer in Egyptians representing 1.6% of the total cancers [2]. Early diagnosis is thew need of the hour as even after surgical exploration of GC patients, local and distant failures were recorded after 6 months [3].

There are 3 pathways that are involved in most GC oncogenesis: the proliferation/stem cell, NF-kappaβ, and Wnt/beta-catenin pathways. The interplay between these pathways may affect disease prognosis and cancer mortality risk [4].

mRNA is involved in various cancers and is associated with cancer prognosis presenting many mRNA gene signatures associated with the cell cycle or immune signature and can be used in the assessment of the prediction of patient survival [5].

The Pleckstrin homology domain containing S1 (PLEKHS1) gene is located on chromosome 10q25.3, encoding the PLEKHS1 protein whose function is still unknown. PLEKHS1 gene is the second gene after telomerase reverse transcriptase (TERT) gene showing frequent promoter somatic non-coding mutations which could be involved in various cancers [6].

The arylacetamide deacetylase (AADAC) gene is located on chromosome 3q25.1 [7] encoding 399 amino acids protein which is localized in the endoplasmic reticulum, involved in various drugs hydrolysis, and expressed in the pancreas, adrenal glands, liver, and gastrointestinal tract [8].

Microsomal AADAC competes against the activity of cytosolic arylamine N-acetyltransferase, which act as a catalyst in the arylamine and heterocyclic amine carcinogens biotransformation pathways [9].

The Cyclin-dependent kinase inhibitor 3 (CDKN3) gene is located on chromosome mapping 14q22 encoding CDKN3 protein also called cyclin-dependent kinase (Cdk)-associated phosphatase (KAP) belonging to the family of protein phosphatases and has a dual function in cell cycle regulation through binding to CDK2 kinase, and interaction with CDK1 [10]. Furthermore, CDKN3 was reported to be downregulated or upregulated in various cancers [11, 12].

A bioinformatics analysis study on GC found PLEKHS1 as a protective prognostic gene and AADAC as a risky prognostic gene while CDKN3 was found to be tightly correlated with the pathogenesis of GC [13], so the present study intended to experimentally evaluate these findings.

To the best of our knowledge, no previous studies experimentally evaluated PLEKHS1, AADAC, and CDKN3 genes expressions in GC so the present study aimed to do that through assessment of them by qRT-PCR in tumorous gastric tissue samples at different stages and grades and non-tumorous gastric tissue.

## Materials and methods

### Samples and studied groups

Gastric tissue samples whether tumorous or adjacent normal gastric tissues away from the tumor were fresh frozen samples just stored in the pathology department repositories at the time of collecting the samples for the present work. Patient names and identity were kept anonymous but pathology and clinical reports were retrieved for the study after getting written permission from the Head of Pathology Department, Faculty of Medicine, Cairo University. Forty (40) tissue samples were obtained from GC patients who underwent surgical resection of the stomach from January 2019 to June 2019 after getting written informed consent. These

tissue samples were divided into two groups: 20 gastric adenocarcinoma samples as group 1 and 20 nontumorous samples as a control group or group 2. All patients were diagnosed pathologically according to the criteria of the American Joint Committee on Cancer. Histopathological diagnoses were performed by two professional pathologists independently. All the patients in the present study progressed during treatment. Metastasis were found as 14 samples obtained from patients with bone metastasis, 4 samples obtained from patients with bone and brain metastasis, and 2 samples obtained from patients with bone and liver metastasis. The current work was performed according to the guidelines and the principles of the Helsinki Declaration [14] and approved by research ethic committee, Faculty of Medicine, Cairo University (letter number was N-24-2019). The disease characteristics in GC group were showed in Table 1.

## Reverse transcription–quantitative polymerase Chain reaction (RT-qPCR)

Total RNA was extracted from tissue sections embedded in paraffin using the RNeasy FFPE kit (Cat No. 73504, Qiagen, USA) in compliance with the manufacturer's protocol. At 260 nm, the extracted RNA had been quantified by spectrophotometry (JENWAY, USA).

## Primer sequence

According to the GenBank RNA sequences, PCR primers were designed [15]. Optimal primer pair had been selected with optimal conditions as 60–65˚C-melting temperature and 90–200 bp- applicant length. All procedures were performed in triplicate. AADAC, PLEKHS1, and CDKN3 primers were presented as shown in Table 2.

**Real-time quantitative 2-step qRT-PCR using SYBR Green.** The current work used software version 3.1 (Applied Biosystems, USA) of Step One plus a real-time PCR system for the analysis of the studied genes. The annealing temperature was set to be optimal as per the PCR protocol and the selected primers.

**Table 1. The disease characteristics among the gastric carcinoma group.**

| Parameters | Gastric carcinoma group n = 20 (%) |
|---|---|
| Tumor size | |
| <3 cm        n (%) | 12 (60) |
| ≥3 cm        n (%) | 8 (40) |
| Lymph nodes | |
| <3        n (%) | 12 (60) |
| ≥3        n (%) | 8 (40) |
| Tumor grades | |
| G1        n (%) | 2 (10) |
| G2        n (%) | 10 (50) |
| G3        n (%) | 8 (40) |
| Tumor stages | |
| II        n (%) | 12 (60) |
| III        n (%) | 6 (30) |
| IV        n (%) | 2 (10) |
| Metastasis (organ) | |
| ≤1        n (%) | 14 (70) |
| >1        n (%) | 6 (30) |

n; number of cases, %; frequency.

**Table 2. PCR primers.**

| Gene | Sequence | Gene bank accession number |
| --- | --- | --- |
| **PLEKHS1** | Forward primer. 5′– TGCTCTCACAGAAGCCACAG –3′ | NM_182601.1 |
| | Reverse primer. 5′– GGACCGGGTAAGAAACAGGG –3′ | |
| **AADAC** | Forward primer. 5′– CTAGAGACCAAGAAGCGGGAC –3′ | NM_001086.3 |
| | Reverse primer. 5′– GTCCCAGGAGCTCCACAAAT –3′ | |
| **CDKN3** | Forward primer. 5′– AGCCGCCCAGTTCAATACAA –3′ | NM_005192.4 |
| | Reverse primer. 5′– CCTGGAAGAGCACATAAACCG –3′ | |
| *GAPDH* | Forward primer 5′– GATGCTGGTGCTGAGTATGTCG –3′ | XR: 598347.1 |
| | Reverse primer 5′– GTGGTGCAGGATGCATTGCTGA –3′ | |

The housekeeping gene that has been used as a reference gene for gene expression normalization was glyceraldehyde 3-phosphate dehydrogenase (GAPDH). cDNA was generated by using total RNA (5 μg), antisense sequence-specific primer (20 pmol), and AMV reverse transcriptase (0.8 μL), for 1 hr. at 37˚C. The present work used the SYBR® Green method (Applied Biosystems, CA, United States) to assess the relative mRNA expressions. The sample of negative control was prepared with no template cDNA used. The optimum experiment requirements for all selected primers were optimized as annealing temperature was 60˚C, Mater Mix of SYBR Green reaction volume was 25 μL, cDNA (1X/reaction) was 3 μL and each primer volume was 900 nmol/L used in the real time polymerase reaction. The amplification optimum conditions were organized according to the manufacturer's instructions as following: Reverse transcription step (50˚C for 15 min), initial PCR activation step (95˚C for 15 min), denaturation step (95˚C for 20 sec) and annealing/extension step (60˚C for 40 sec) along 40 cycles.

**Calculation of relative gene expression.** The common double delta threshold cycle (ΔΔ Ct) method for relative quantification (RQ) was used for the calculation of the expression of the studied genes by using passive reference dye (ROX) for normalization of the fluorescence and glyceraldehyde-3-phosphate dehydrogenase (GAPDH) as a reference gene. Applied Biosystems Step One plus software was used to calculate the Ct values of the reference gene and the analyzed genes. The Ct values of the target genes and the housekeeping gene (GAPDH) were included in PCR data analysis. No template cDNA used in the negative control sample. All values were normalized to GAPDH and expressed as fold changes relative to its control samples' background levels.

RQ was calculated according to the following equation: [16].

Δ Ct (in GC) sample = Ct studied gene–Ct reference gene

Δ Ct (in control sample) = Ct studied gene–Ct reference gene

Δ Δ Ct = Δ Ct (in GC) sample–Δ Ct (in control sample)

$RQ = 2^{-(\Delta \Delta Ct)}$

## Statistical analysis

Data was analyzed using the Windows 10 version 25 Statistical Package of Social Science (SPSS) applications. MedCalc® statistical software was used for production of the Receiving Operating Characteristic (ROC) curve. Easy descriptive analysis was done in the form of numbers and percentages of qualitative data, and arithmetic mean as central tendency measurement and standard deviations as a calculation of the dispersion of quantitative parametric

data. Quantitative data used in the analysis were first tested for normality by a one-sample Kolmogorov-Smirnov test in each study group and then selected for inferential statistical tests. For quantitative parametric data: independent t-test samples, used to compare quantitative measurements between two independent classes. For qualitative results, the Chi-square test was used to compare two qualitative groups out of more than two. The P-value< 0.05 was statistically significant. ROC curve was used for prediction of the gastric adenocarcinoma and its metastatic susceptibility using gene expressions of AADAC CDKN3 and PLEKHS1. Sensitivity, and specificity were calculated at the optimal cut off values of the studied genes.

## Results

### PLEKHS1, AADAC, and CDKN3 gene expression levels in the studied groups

There were high significant (p<0.001) elevations in AADAC gene expression and CDKN3 gene expression in the gastric carcinoma group expressed by (M±SD) [(0.89±0.124), and (0.80 ±0.09)] compared to the corresponding values in the control group [(0.41±0.04), and (0.43 ±0.126) respectively] while PLEKHS1 gene expression showed a high significant (p<0.001) decrease in the gastric carcinoma group (1.68±0.31) compared to the corresponding values in the control group (2.70±0.17) as shown in Fig 1.

### Relations between the tumor progression and the studied parameters

AADAC gene expression showed high significant (p<0.001) increases in patients with tumor progression signs (tumor size ≥3 cm, ≥3 lymph nodes, higher grades, and higher stages) and a significant (p<0.05) increase in patients with high metastasis (> one organ) while PLEKHS1 and CDKN3 gene expressions showed no significant differences in the same patients compared to those with less tumor progression signs as shown in Table 3.

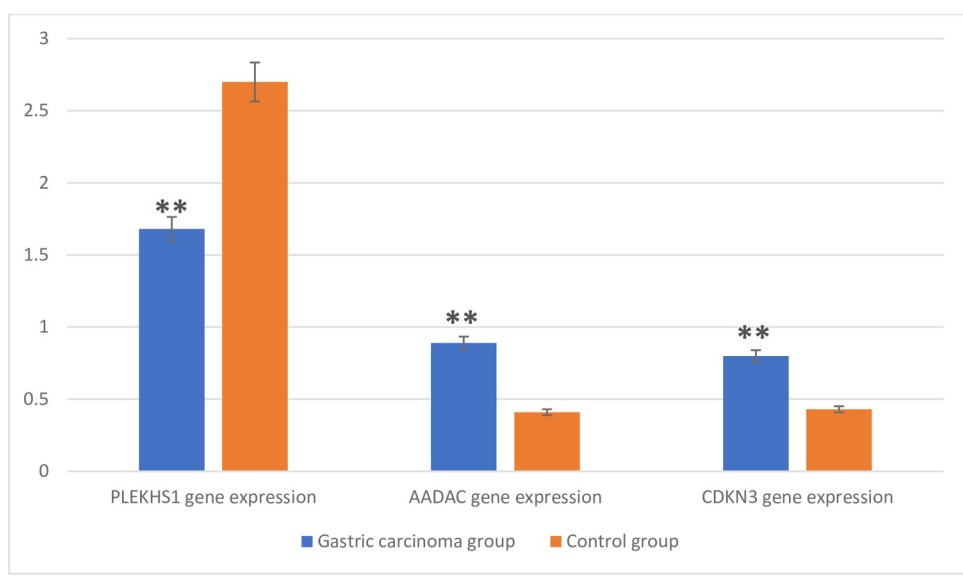

**Fig 1. Comparison of PLEKHS1, AADAC, CDKN3 gene expressions between the studied groups.** High significant elevations in AADAC gene expression and CDKN3 gene expression in the gastric carcinoma group compared to the control group while PLEKHS1 gene expression showed a high significant decrease in the gastric carcinoma group compared to the control group. **: statistically high significant differences (P<0.001) compared to the corresponding values in control group.

**Table 3. Comparison between the levels of the studied parameters in different disease characteristics among the gastric carcinoma group.**

| Parameters | PLEKHS1 gene expression | AADAC gene expression | CDKN3 gene expression |
|---|---|---|---|
| **Tumor size** | | | |
| **<3 cm** | 1.78±0.33 | 0.82±0.09 | 0.8±0.08 |
| **≥3 cm** | 1.54±0.25 | 1.00±0.04 | 0.8±0.11 |
| **p-value** | 0.098 | <0.001** | 0.999 |
| **Lymph nodes** | | | |
| **<3** | 1.78±0.33 | 0.82±0.09 | 0.8±0.08 |
| **≥3** | 1.54±0.25 | 1.00±0.04 | 0.8±0.11 |
| **p-value** | 0.098 | <0.001** | 0.999 |
| **Tumor grades** | | | |
| **G1 & G2** | 1.78±0.33 | 0.82±0.09 | 0.8±0.08 |
| **G3** | 1.54±0.25 | 1.00±0.04 | 0.8±0.11 |
| **p-value** | 0.098 | <0.001** | 0.999 |
| **Tumor stages** | | | |
| **II** | 1.78±0.33 | 0.81±0.09 | 0.8±0.08528 |
| **III** | 1.46±0.24 | 1.00±0.05! | 0.83±0.1 |
| **IV** | 1.77±0 | 1.01±0! | 0.7±0 |
| **p-value** | 0.125 | <0.001** | 0.212 |
| **Metastasis** | | | |
| **(organ)** | 1.75±0.32 | 0.85±0.12354 | 0.81±0.08 |
| **≤1** | 1.54±0.29 | 0.99±0.03 | 0.77±0.10 |
| **>1** | 0.172 | 0.013* | 0.300 |
| **p-value** | | | |

PLEKHS1; pleckstrin homology domain containing S1, AADAC; arylacetamide deacetylase, CDKN3; cyclin dependent kinase inhibitor 3

**; statistically high significant difference (P<0.001).

*; statistically significant difference (P<0. 05). G; grade of cancer.

## Significant correlations in the study

There were highly significant positive correlations between AADAC gene expression and both tumor grades (r = .603, p<0.05) and tumor stages (r = .756, p<0.001) and there was a significant negative correlation between PLEKHS1 gene expression and tumor grades (r = -.462, p<0.05) while CDKN3 gene expressions had no significant correlations with tumor grades and stages. A high significant negative correlation between AADAC, and CDKN3 gene expression (r = -.760, p<0.001) was found.

**ROC curve analysis of the studied parameters in the prediction of the presence of gastric adenocarcinoma.** ROC curve analysis of the three studied parameters in the prediction of the presence of gastric adenocarcinoma indicated that the optimum cut-off value for PLEKHS1gene expression was ≤2.21 with AUC 0.993, 95% sensitivity and 100% specificity, for AADAC gene expression, a cut-off value of >0.48 with AUC 0.990, 95% sensitivity and 100% specificity, and for CDKN3 gene expression, a cutoff value of >0.65 with AUC 0.990, 100% sensitivity and 95% specificity as shown in Fig 2.

## Discussion

The World Health Organization estimates that in 2018, GC accounted for 783,000 deaths worldwide [17]. mRNA dysfunction has been shown to be involved in various cancers and has been significantly associated with cancer prognosis [5].

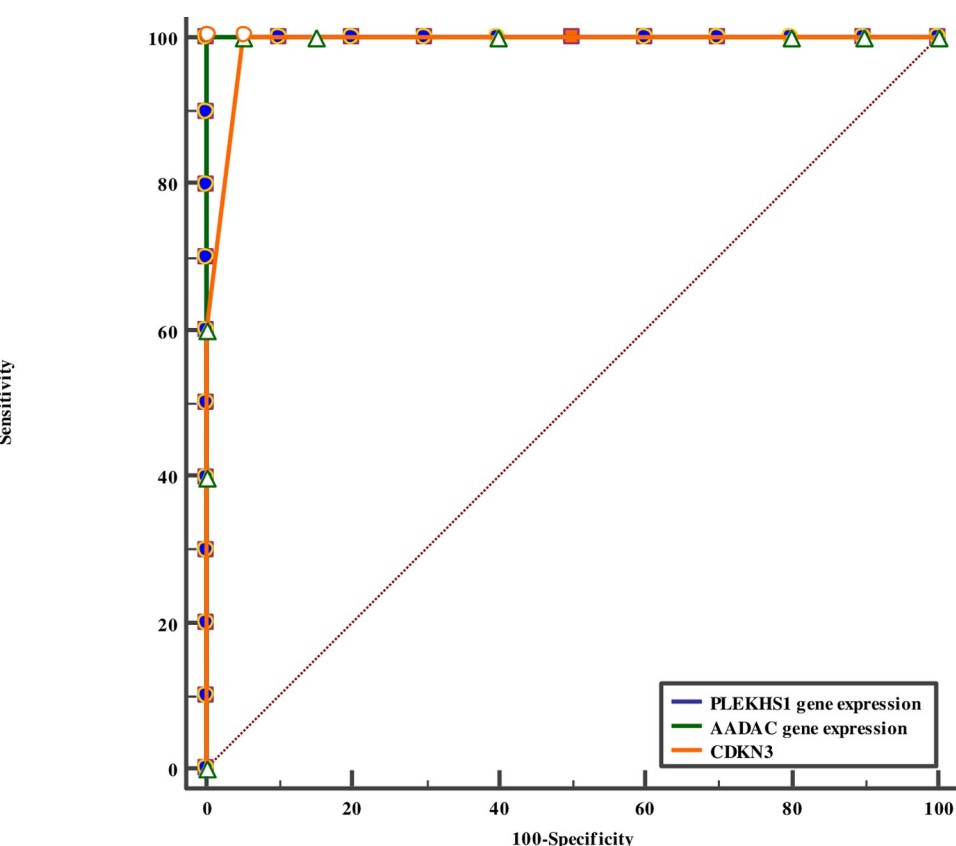

**Fig 2. Comparison between PLEKHS1, AADAC, CDKN3 gene expressions in the prediction of the presence of gastric adenocarcinoma.** PLEKHS1, AADAC, CDKN3 gene expressions showed high significant sensitivity and specificity in the prediction of the presence of GC.

The current study aimed to evaluate PLEKHS1, AADAC and CDKN3 gene expressions in GC because they were included in gene-signature for GC in some bioinformatics analysis studies [13, 18–20].

The present study showed a highly significant (p<0.001) decrease in PLEKHS1 gene expression levels in the GC group as compared to the control group. Also, PLEKHS1 gene expression showed high significant sensitivity (95%) and specificity (100%) in the prediction of the presence of GC. Additionally, a significant negative correlation between PLEKHS1 gene expression and tumor grades was found pointing to its probable protective role in GC.

Liu et al. [13] coincided with these results because they reported PLEKHS1as a protective prognostic gene in GC using multiple gene expression profile datasets integrated analysis.

These findings coincided with Chen et al. [18], who reported PLEKHS1 as one of the four genes of the prognostic-protective marker for predicting the overall survival (OS) of GC using univariate, multivariate, and Lasso Cox regression analyses but Zhang et al. [21] showed PLEKHS1 upregulation in the intestinal-type-GC and it was considered as one of a five-gene signature for the disease using The Cancer Genome Atlas (TCGA) database.

Coinciding with the present findings Zhou et al. [22] by using the TCGA database, showed that PLEKHS1 was downregulated in stage I Testicular germ cell tumors (TGCTs) patients, and it was reported as one of the new eight-gene signatures that could be used in the prediction of relapse-free survival (RFS) having a protective role in TGCT relapse.

On the contrary, some bioinformatics analysis studies reported the upregulation of PLEKHS1 in some cancers from these studies; Xiong et al. [23] who reported that PLEKHS1 was one of the top 10 up-regulated genes which were involved in the 5-5-Fluorouracil-based chemoradiation resistance in colorectal cancer (CRC), Yong et al. [24] who showed that PLEKHS1 among the upregulated genes in colon adenocarcinoma (COAD), Zhenfeng et al. [25] who reported that PLEKHS1 as one of the differentially expressed genes (DEGs) which upregulated and associated with poor prognosis in hepatocellular carcinoma (HCC) and Pignot et al. [26] who showed PLEKHS1 upregulation in non-muscle-invasive bladder cancer (NMIBC).

These results could be explained by the previous studies reporting that PLEKHS1 proteins could cause noncoding mutations of upstream and promoter elements, which could be involved in tumorigenesis [6]. According to a genome-wide analysis, mutations in non-coding regions of PLEKHS1 were found in 14 types of cancers and are associated with poor prognosis [27].

The present study showed a highly significant ($p < 0.001$) increase in AADAC gene expression levels in the GC group as compared to the control group. AADAC gene expression showed high significant ($p < 0.001$) increases in patients with tumor progression signs (tumor size $\geq 3$ cm, $\geq 3$ lymph nodes, higher grades, and higher stages) and a significant ($p < 0.05$) increase in patients with high metastasis (> one organ). There were highly significant positive correlations between AADAC gene expression and both tumor grades and, tumor stages. Also, AADAC gene expression showed high significant sensitivity (95%) and specificity (100%) in the prediction of the presence of GC.

These findings matched with the findings of Wu et al. [19] who reported AADAC upregulation, and it was one of the nine-gene prognostic signature for gastric carcinoma using integrated bioinformatics analyses.

Wu et al. [28] coincided with the present results as they found the AADAC gene to be one of five survival-related genes significantly upregulated and significantly correlated with overall survival (OS) in resectable pancreatic cancer (PC) by using the TCGA database followed by multivariant analysis. They explained their results as the protein coded by AADAC protein is extensively implicated in the hydrolysis of various drugs [9], whose function may be related to chemotherapy resistance in pancreatic cancer.

The present study showed a highly significant ($p < 0.001$) increase in CDKN3 gene expression levels in the GC group as compared to the control group. Also, CDKN3 gene expression showed high significant sensitivity (100%) and specificity (95%) in the prediction of the presence of GC. CDKN3 gene expressions showed no significant correlations with tumor grades and stages.

These results coincided with Li et al. [20] who showed knockdown of CDKN3 in human GC cells, significantly decreased cancer cell proliferation, invasion, migration, and adhesion leading to cell apoptosis while upregulation of CDKN3 in GC was significantly associated with bad prognosis suggesting that CDKN3 could be a novel therapeutic target in GC treatment.

Liu et al. [29] agreed with the current findings as they showed that CDKN3 loss extensively decreased cancer cell proliferation and significantly decreased the invasion and adhesion abilities and CDKN3 loss had been associated with downregulation of CDK1 and CDK2.

Some studies coincided with the current findings but in other gastrointestinal cancers. From these studies, Wang et al. [30] showed that CDKN3 was upregulated in esophageal cancer and considered as a prognostic marker of the disease using bioinformatic analysis which showed the association between CDKN3 and the cell cycle stage transition, DNA replication, and DNA repair system, Further, Yang et al. [31] showed that CDKN3 was upregulated in colorectal cancer.

The mechanism of CDKN3 upregulation could be explained by the hypomethylation of its promoter region [32].

On the contrary, CDKN3 was reported as a tumor suppressor in some cancers [11] as it is a negative regulator of Cyclin-dependent kinases (CDK1 and CDK2) [33] but the exact effect of CDKN3 in cancer cell proliferation, either enhancing or precluding, it cannot be explained by its CDK regulation, alone.

In the present study, a high significant negative correlation between AADAC, and CDKN3 gene expression was found. This relation may be found as AADAC acts as an oncogene while CDKN3 acts as a tumor suppressor gene, whatever this finding needs more further future studies.

## Conclusion

In conclusion, AADAC and CDKN3 genes were significantly upregulated in gastric adenocarcinoma. On the other hand, PLEKHS1 was significantly downregulated in gastric adenocarcinoma and showed an inverse correlation with tumor grade suggesting its protective role in GC. Finally, AADAC, CDKN3, and PLEKHS1 could be used as diagnostic and predictive biomarkers of GC, and AADAC could be used as a prognostic marker in these patients. These genes are suggested to be used as novel therapeutic targets in GC for further future confirming studies on larger samples.

## Supporting information

**S1 Data.**
(SAV)

## Acknowledgments

The current study was supported by Head of Pathology Department Faculty of Medicine Cairo University who provided great facilitation during collection of archival gastric tissue samples. We express our appreciation of the cooperation of Prof. Dr. Soliman Saba, Head of Pathology Department.

## Author Contributions

**Data curation:** Asmaa M. Othman.

**Formal analysis:** Marwa Sayed Abdel-Tawab.

**Investigation:** Ragaey A. Eid.

**Methodology:** Hanan Fouad.

**Supervision:** Ahmed Hassan.

**Writing – original draft:** Marwa Sayed Abdel-Tawab.

**Writing – review & editing:** Marwa Abdeltawab Mohammed, Hoda Ramadan Reyad.

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
