## [Decision Letter · Decision Letter 0]

16 Nov 2021

PONE-D-21-19753Evaluation of Gene expression of PLEKHS1, AADAC, and CDKN3 as novel genomic markers in  Gastric Carcinoma in Egyptian PatientsPLOS ONE

Dear Dr. Marwa Sayed Abdel-Tawab,

Thank you for submitting your manuscript to PLOS ONE. After careful consideration, we feel that it has merit but does not fully meet PLOS ONE’s publication criteria as it currently stands. Therefore, we invite you to submit a revised version of the manuscript that addresses the points raised during the review process.

We look forward to receiving your revised manuscript.

Kind regards,

Wen-Wei Sung, M.D., Ph.D.

Academic Editor

PLOS ONE

Journal Requirements:

Reviewers' comments:

Reviewer's Responses to Questions

**Comments to the Author**

1. Is the manuscript technically sound, and do the data support the conclusions?

Reviewer #1: No

Reviewer #2: Yes

Reviewer #3: Partly

Reviewer #4: Partly

2. Has the statistical analysis been performed appropriately and rigorously? 

Reviewer #1: Yes

Reviewer #2: Yes

Reviewer #3: Yes

Reviewer #4: Yes

3. Have the authors made all data underlying the findings in their manuscript fully available?

Reviewer #1: Yes

Reviewer #2: Yes

Reviewer #3: No

Reviewer #4: Yes

4. Is the manuscript presented in an intelligible fashion and written in standard English?

Reviewer #1: No

Reviewer #2: Yes

Reviewer #3: No

Reviewer #4: Yes

5. Review Comments to the Author

Reviewer #1: Abdel-Tawab et al demonstrated in this manuscript that PLEKHS1, AADAC and CDKN3 can act as markers for gastric carcinoma based on clinical samples from Egyptian patients. While the authors did show the correlation of gastric carcinoma with mRNA expression of indicated genes, that is the sole result in this study, and there are many questions regarding PLEKHS1, AADAC, and CDKN3 in gastric carcinoma to be answered.

1. The rationales for choosing PLEKHS1, AADAC and CDKN3 as targets are missing in introduction or discussion.

2. By emphasizing Egyptian patients in the study, are there any difference for the expression profile of PLEKHS1, AADAC and CDKN3 between Egyptian patients and patient from other area?

3. The association between PLEKHS1, AADAC, and CDKN3 are missing. Moreover, the molecular mechanism for the impacts of PLEKHS1, AADAC, and CDKN3 in tumor is not showed.

Reviewer #2: In my opinion, the authors performed a well-constructed study to test their research questions. The statistical analysis seemed to me to be adequate and well carried out. In the discussion, the authors made a careful analysis of their results and compared them with other works already published.

Reviewer #3: Comments:

In the present report the authors have tried to experimentally validate PLEKHS1, AADAC, and CDKN3 genes expressions in gastric cancer. A previous study by Liu X et al 2018 using integrated analysis of multiple gene expression profile datasets had narrowed down to a list comprising of the above said markers. The authors have a valid aim in further validating the markers in an independent set of samples. The methodology and the statistical analysis adopted for the study is appropriate. However, their validation process has several deficiencies due to which in the present form cannot be recommended for publication. The following are the major short comings.

1. The authors need to provide a clear definition of their sample set like inclusion and exclusion criteria. For instance, they need to provide information on the source of their normal samples. Paired normal (Sourced from adjacent normal tissues away from the tumor) or normal gastric tissue. It would be ideal if they could compare with both kinds of normal tissue. Histological subtype of the samples assessed. The overall distribution of samples across the various stages is highly limited. For instance, only 2 samples are used in stage IV. In table 1 it is not clear what the authors mean the by the term “Outcome”. The row on Metastasis needs to describe which organ. Visceral organ metastasis is considered stage IV disease.

2. The tissue source is from archival FFPE sections, though routinely used for analysis there are some drawbacks regarding the quality of RNA sourced from this source. Preferably including fresh frozen samples in the study will help in overcome questions regarding the quality of the assays performed.

3. Additional protein expression studies using Immunohistochemistry and archival FFPE sections for the markers will considerably increase the impact of the findings by increasing the sample numbers.

4. In the results section the authors mention fold change values “[(0.89±0.124), and (0.80±0.09)] compared to the corresponding values in the control group [(0.41±0.04), and (0.43±0.126) respectively] “ what the common calibrator was used to determine the fold changes in tumor and control groups?

5. It is not clear why the authors mention “Demographic Data in the studied groups:” considering the fact that the control samples were sourced from the same gastric tumor tissue (paired normal) else the authors need to specifically mention this.

6. While mentioning data on “Relations between the tumor progression and the studied parameters” it is necessary to give back ground information on how many of the patients progressed during treatment and how many did not do so.

7. Overall, considering that the present study is a validation analysis a larger sample set is required to assess the markers along with clinicopathological features such as H. Pylori status, subtypes of gastric cancer.

Reviewer #4: This study suggests that PLAKHS1, AADAC and CDKN3 genes can be used as biomarkers and prognostic marker (in the caso of AADAC gene) of gastric cancer. This topic is of great clinical relevance. However, the manuscript should be improved before publication. Initially, English should be revised and the tittle "Figure Ligands" must be corrected to "Figure legends". These, should also be improved and include more descriptive data on the figures to which they refer. The style of the References must be corrected and adapted to the standard recommended by the Journal.

With respect to the Subjects and Methods section, the number of specimens analyzed in some groups is low (e.g. n=2 for G1 tumor grade and tumor stage IV), unfortunately; only 1 housekeeping gene was used (ideally authors should work with 2 housekeeping genes) and another technique that allows the assessment of proteins that are encoded by the analyzed genes (such as IHC) should be added in order to strengthen the results.

6. PLOS authors have the option to publish the peer review history of their article (what does this mean?). If published, this will include your full peer review and any attached files.

Reviewer #1: No

Reviewer #2: No

Reviewer #3: No

Reviewer #4: No

---

## [Author Response · Author response to Decision Letter 0]

11 Dec 2021

Reviewer #1

1. The rationales for choosing PLEKHS1, AADAC and CDKN3 as targets are missing in introduction or discussion.

Response: 

The rationales for choosing PLEKHS1, AADAC and CDKN3 as targets are included in the following paragraph in the introduction “A bioinformatics analysis study on GC found PLEKHS1 as a protective prognostic gene and AADAC as a risky prognostic gene while CDKN3 was found to be tightly correlated with the pathogenesis of GC [13], so the present study intended to experimentally evaluate these findings.” 

2. By emphasizing Egyptian patients in the study, are there any difference for the expression profile of PLEKHS1, AADAC and CDKN3 between Egyptian patients and patient from other area?

Response: 

We do not know if there are any difference for the expression profile of PLEKHS1, AADAC and CDKN3 between Egyptian patients and patient from other area so, we delete these words “in Egyptian patients “from the title.

3. The association between PLEKHS1, AADAC, and CDKN3 are missing. Moreover, the molecular mechanism for the impacts of PLEKHS1, AADAC, and CDKN3 in tumor is not showed.

Response: 

A high significant negative correlation between AADAC, and CDKN3 gene expression (r= -.760, p<0.001) was found. This paragraph was added to the abstract, results, and discussion. The molecular mechanism for the impacts of PLEKHS1, AADAC, and CDKN3 in tumor needs more evaluating further future studies as we mentioned at the end of the abstract and conclusion.

Reviewer #3

1. The authors need to provide a clear definition of their sample set like inclusion and exclusion criteria. For instance, they need to provide information on the source of their normal samples. Paired normal (Sourced from adjacent normal tissues away from the tumor) or normal gastric tissue. It would be ideal if they could compare with both kinds of normal tissue. Histological subtype of the samples assessed. The overall distribution of samples across the various stages is highly limited. For instance, only 2 samples are used in stage IV. In table 1 it is not clear what the authors mean the by the term “Outcome”. The row on Metastasis needs to describe which organ. Visceral organ metastasis is considered stage IV disease.

Response: 

We mentioned that the source of the normal samples were adjacent normal tissues away from the tumor.

We mean by outcome in table 1; tissues have tumor (positive) or not (negative) and we delete this item from the table as if it could not be understood. 

only 2 samples are used in stage IV because we were eager to have fresh frozen samples but at this time, only 2 samples in stage IV were available.

Metastasis were found as 14 samples obtained from patients with bone metastasis, 4 samples obtained from patients with bone and brain metastasis, and 2 samples obtained from patients with bone and liver metastasis

2. The tissue source is from archival FFPE sections, though routinely used for analysis there are some drawbacks regarding the quality of RNA sourced from this source. Preferably including fresh frozen samples in the study will help in overcome questions regarding the quality of the assays performed.

Response:

The samples already were fresh frozen samples just stored in the pathology department repositories at the time of collecting the samples for the present work.

3. Additional protein expression studies using Immunohistochemistry and archival FFPE sections for the markers will considerably increase the impact of the findings by increasing the sample numbers.

Response: These additional protein expression studies need more cost, and we have no any funding source except our own work, so we add that our findings need more evaluating further future studies to overcome all limitations in our study.

4. In the results section the authors mention fold change values “[(0.89±0.124), and (0.80±0.09)] compared to the corresponding values in the control group [(0.41±0.04), and (0.43±0.126) respectively] “what the common calibrator was used to determine the fold changes in tumor and control groups?

Response: The common calibrator was used to determine the fold changes in tumor and control groups expressed by (M±SD), these words are added to the result section.

5. It is not clear why the authors mention “Demographic Data in the studied groups:” considering the fact that the control samples were sourced from the same gastric tumor tissue (paired normal) else the authors need to specifically mention this.

Response: We mean demographic data in the studied subjects not groups. We delete this paragraph to avoid misunderstanding from readers.

6. While mentioning data on “Relations between the tumor progression and the studied parameters” it is necessary to give background information on how many of the patients progressed during treatment and how many did not do so.

Response: All the patients in our study progressed during treatment. We add these words to the subject and methods section.

7. Overall, considering that the present study is a validation analysis a larger sample set is required to assess the markers along with clinicopathological features such as H. Pylori status, subtypes of gastric cancer.

Response:

You are right! this is a one of our study limitations, but these additional data were not available for all patients, so we could not do this assessment. 

Reviewer #4

1. Initially, English should be revised, and the tittle "Figure Ligands" must be corrected to "Figure legends". 

Response: It is done 

2. These should also be improved and include more descriptive data on the figures to which they refer. 

Response: It is done

3. The style of the References must be corrected and adapted to the standard recommended by the Journal.

Response: It is done

4. With respect to the Subjects and Methods section, the number of specimens analyzed in some groups is low (e.g., n=2 for G1 tumor grade and tumor stage IV)

Response:

only 2 samples are used in stage IV or in grade 1 because we were eager to have fresh frozen samples but during the period of our samples collection, only theses samples were available. 

5. Unfortunately, only 1 housekeeping gene was used (ideally authors should work with 2 housekeeping genes) and another technique that allows the assessment of proteins that are encoded by the analyzed genes (such as IHC) should be added in order to strengthen the results.

Response:

These additional protein expression studies need more cost, and we have no any funding source except our own work, so we add that our findings need more evaluating further future studies to overcome all limitations in our study.

---

## [Decision Letter · Decision Letter 1]

30 Dec 2021

PONE-D-21-19753R1Evaluation of Gene expression of PLEKHS1, AADAC, and CDKN3 as novel genomic markers in  Gastric CarcinomaPLOS ONE

Dear Dr. Marwa Sayed Abdel-Tawab,

Thank you for submitting your manuscript to PLOS ONE. After careful consideration, we feel that it has merit but does not fully meet PLOS ONE’s publication criteria as it currently stands. Therefore, we invite you to submit a revised version of the manuscript that addresses the points raised during the review process.

There are still major concerns need further revision according to the reports from reviewer 4. Please try to revise the manuscript with more supportive results.

We look forward to receiving your revised manuscript.

Kind regards,

Wen-Wei Sung, M.D., Ph.D.

Academic Editor

PLOS ONE

Reviewers' comments:

Reviewer's Responses to Questions

**Comments to the Author**

1. If the authors have adequately addressed your comments raised in a previous round of review and you feel that this manuscript is now acceptable for publication, you may indicate that here to bypass the “Comments to the Author” section, enter your conflict of interest statement in the “Confidential to Editor” section, and submit your "Accept" recommendation.

Reviewer #3: All comments have been addressed

Reviewer #4: (No Response)

2. Is the manuscript technically sound, and do the data support the conclusions?

Reviewer #3: Yes

Reviewer #4: Partly

3. Has the statistical analysis been performed appropriately and rigorously? 

Reviewer #3: Yes

Reviewer #4: Yes

4. Have the authors made all data underlying the findings in their manuscript fully available?

Reviewer #3: Yes

Reviewer #4: Yes

5. Is the manuscript presented in an intelligible fashion and written in standard English?

Reviewer #3: No

Reviewer #4: No

6. Review Comments to the Author

Reviewer #3: The authors have addressed the queries raised by the reviewer. The study size is small however considering that additional samples will be required a fact that has been acknowledged in the authors response the manuscript can be accepted.

Minor points

Correction for English language usage may be necessary to improve the readability of the paper.

Reviewer #4: Although the authors made modification to the original text, not all the suggested corrections ans questions were properly performed or answered.

7. PLOS authors have the option to publish the peer review history of their article (what does this mean?). If published, this will include your full peer review and any attached files.

Reviewer #3: No

Reviewer #4: No

---

## [Author Response · Author response to Decision Letter 1]

31 Dec 2021

Response to reviewers

Reviewer #4

1. Initially, English should be revised, and the tittle "Figure Ligands" must be corrected to "Figure legends". 

Response: It is done 

2. These should also be improved and include more descriptive data on the figures to which they refer. 

Response: It is done

3. The style of the References must be corrected and adapted to the standard recommended by the Journal.

Response: It is done

4. With respect to the Subjects and Methods section, the number of specimens analyzed in some groups is low (e.g., n=2 for G1 tumor grade and tumor stage IV)

Response:

GC has low incidence and prevalence in Egyptians representing 1.6% of the total cancers [1], so we could not find more fresh frozen samples during the period of our samples collection. We were eager to work as rapid as possible to ensure the stability of RNA in the collected samples only 2 samples were used in stage IV or in grade 1 because, only theses samples were available in the fresh frozen samples during the period of our samples collection.

5. Unfortunately, only 1 housekeeping gene was used (ideally authors should work with 2 housekeeping genes) and another technique that allows the assessment of proteins that are encoded by the analyzed genes (such as IHC) should be added in order to strengthen the results.

Response:

Using of one housekeeping gene is the routine method at our molecular biology unit of Kasr Alaini Medicine College in Egypt. This method is accepted in many published articles introduced by researchers to international journals and PLoS One journal [2].

These additional protein expression studies need more cost, and we have no any funding source except our own work, so our study evaluates the gene expression ant it is obvious in the title of the article mentioning that that our findings need more confirming further future studies to overcome all limitations in our study. It is very difficult to do additional work on the same samples. 

References 

1.Abdelwahab Hashem TA, El-Fotouh MA, Ehab A, El Rebey HS, Satar MA, Attallah HS. Her-2 neu status in gastric carcinoma in Egyptian patients: The epidemiology and the response to chemotherapy. Menoufia Med J 2016; 29: 449-53.

2. Shan L, Lian F, Guo L, Qiu T, Ling Y, Ying J, Lin D. Detection of ROS1 gene rearrangement in lung adenocarcinoma: comparison of IHC, FISH and real-time RT-PCR. PLoS One. 2015 Mar 5;10(3):e0120422. doi: 10.1371/journal.pone.0120422. PMID: 25742289; PMCID: PMC4351102.

---

## [Decision Letter · Decision Letter 2]

17 Jan 2022

PONE-D-21-19753R2Evaluation of Gene expression of PLEKHS1, AADAC, and CDKN3 as novel genomic markers in  Gastric CarcinomaPLOS ONE

Dear Dr. Marwa Sayed Abdel-Tawab,

Thank you for submitting your manuscript to PLOS ONE. After careful consideration, we feel that it has merit but does not fully meet PLOS ONE’s publication criteria as it currently stands. Therefore, we invite you to submit a revised version of the manuscript that addresses the points raised during the review process.

The reviewers have major concerns include, but not limited, the references. Please revise the manuscript carefully.

We look forward to receiving your revised manuscript.

Kind regards,

Wen-Wei Sung, M.D., Ph.D.

Academic Editor

PLOS ONE

Reviewers' comments:

Reviewer's Responses to Questions

**Comments to the Author**

1. If the authors have adequately addressed your comments raised in a previous round of review and you feel that this manuscript is now acceptable for publication, you may indicate that here to bypass the “Comments to the Author” section, enter your conflict of interest statement in the “Confidential to Editor” section, and submit your "Accept" recommendation.

Reviewer #3: All comments have been addressed

Reviewer #4: (No Response)

2. Is the manuscript technically sound, and do the data support the conclusions?

Reviewer #3: Yes

Reviewer #4: Partly

3. Has the statistical analysis been performed appropriately and rigorously? 

Reviewer #3: Yes

Reviewer #4: Yes

4. Have the authors made all data underlying the findings in their manuscript fully available?

Reviewer #3: Yes

Reviewer #4: Yes

5. Is the manuscript presented in an intelligible fashion and written in standard English?

Reviewer #3: Yes

Reviewer #4: Yes

6. Review Comments to the Author

Reviewer #3: (No Response)

Reviewer #4: Unfortunatelly, the References' section remains wrong, with several mistakes, which demonstrates the authors' regrettable lack of attention.

7. PLOS authors have the option to publish the peer review history of their article (what does this mean?). If published, this will include your full peer review and any attached files.

Reviewer #3: No

Reviewer #4: No

---

## [Author Response · Author response to Decision Letter 2]

20 Jan 2022

Reviewer #4

1. The style of the References must be corrected and adapted to the standard recommended by the Journal.

Response: I reformatted, revised all references through EndNote X9 and corrected all mistakes in the references.

---

## [Decision Letter · Decision Letter 3]

28 Feb 2022

Evaluation of Gene expression of PLEKHS1, AADAC, and CDKN3 as novel genomic markers in  Gastric Carcinoma

PONE-D-21-19753R3

Dear Dr. Marwa Sayed Abdel-Tawab,

We’re pleased to inform you that your manuscript has been judged scientifically suitable for publication and will be formally accepted for publication once it meets all outstanding technical requirements.

Kind regards,

Wen-Wei Sung, M.D., Ph.D.

Academic Editor

PLOS ONE

Reviewers' comments:

Reviewer's Responses to Questions

**Comments to the Author**

1. If the authors have adequately addressed your comments raised in a previous round of review and you feel that this manuscript is now acceptable for publication, you may indicate that here to bypass the “Comments to the Author” section, enter your conflict of interest statement in the “Confidential to Editor” section, and submit your "Accept" recommendation.

Reviewer #4: All comments have been addressed

2. Is the manuscript technically sound, and do the data support the conclusions?

Reviewer #4: Yes

3. Has the statistical analysis been performed appropriately and rigorously? 

Reviewer #4: Yes

4. Have the authors made all data underlying the findings in their manuscript fully available?

Reviewer #4: Yes

5. Is the manuscript presented in an intelligible fashion and written in standard English?

Reviewer #4: Yes

6. Review Comments to the Author

Reviewer #4: (No Response)

7. PLOS authors have the option to publish the peer review history of their article (what does this mean?). If published, this will include your full peer review and any attached files.

Reviewer #4: No

---

## [Editor Report · Acceptance letter]

2 Apr 2022

PONE-D-21-19753R3 

Evaluation of Gene expression of PLEKHS1, AADAC, and CDKN3 as novel genomic markers in Gastric Carcinoma 

Dear Dr. Abdel-Tawab:

I'm pleased to inform you that your manuscript has been deemed suitable for publication in PLOS ONE. Congratulations! Your manuscript is now with our production department. 

Kind regards, 

on behalf of

Dr. Wen-Wei Sung 

Academic Editor

PLOS ONE